# Functional Characterization of a New Salt Stress Response Gene, *PeCBL4*, in *Populus euphratica* Oliv

Meiqiao Qu [1,†], Qi Sun [1,†], Ningning Chen [1], Zhuoyan Chen [1], Hechen Zhang [2], Fuling Lv [3,*] and Yi An [1,*]

1 State Key Laboratory of Subtropical Silviculture, College of Forestry and Biotechnology, Zhejiang A&F University, Hangzhou 311300, China; q17781090492@163.com (M.Q.); sunqi822@zafu.edu.cn (Q.S.); 20220152@zafu.edu.cn (N.C.)

2 Horticultural Research Institute, Henan Academy of Agricultural Sciences, No. 116, Huayuan Road, Jinshui District, Zhengzhou 450002, China

3 Chinese Academy of Forestry, Beijing 100091, China

* Correspondence: lflshiplmm@126.com (F.L.); anyi@zafu.edu.cn (Y.A.)

† These authors contributed equally to this work.

**Abstract:** *Populus euphratica* is a typical stress-resistant tree species that provides valuable natural genetic resources for breeding salt-tolerant plants. The calcineurin B-like (CBL)-interacting protein kinase (CIPK) network plays an important role in regulating plant responses to abiotic stresses. The aim of this study was to characterize the function of a new CBL member, *PeCBL4*, in response to abiotic stresses. *PeCBL4* was cloned, and sequence analysis was performed. The subcellular localization of PeCBL4 was determined using the fusion expression vector of GFP. Yeast two-hybrid assays and bimolecular fluorescence complementation were performed to identify PeCIPK members that interacted with PeCBL4. *PeCBL4* was then transformed into the corresponding *Arabidopsis thaliana* mutants. $Na^+$ and $K^+$ content as well as their net fluxes were determined under high salt stress and low $K^+$ stress. Phylogenetic tree analysis showed that *PeCBL4* was clustered together with *PtCBL4* and belonged to the same subgroup as *AtCBL4*. Subcellular localization indicated that PeCBL4 was expressed on the plasma membrane. Yeast two-hybrid assays and bimolecular fluorescence complementation showed that PeCBL4 interacted with PeCIPK24 and PeCIPK26. In addition, under high salt stress, the $Na^+$ efflux capacities of seedlings decreased in *sos3* mutants, and transgenic plants of *PeCBL4* enhanced efflux capacities. In addition, the overexpression of *PeCBL4* negatively influenced the influx capacity of $K^+$. PeCBL4 interacts with PeCIPK24 and PeCIPK26 and regulates $Na^+/K^+$ balance under low $K^+$ and high salt stress.

**Keywords:** *Populus euphratica*; salt stress; potassium stress; calcineurin B-like protein





## 1. Introduction

The forest ecosystem is a major component of the terrestrial ecosystems in the biosphere, and excessive soil salinity is a critical issue worldwide because it can have adverse effects on forest and agricultural plant sustainability and productivity [1,2]. It is estimated that approximately 10% of arable land and 25% to 30% of irrigated land are affected by salinity or sodicity [3]. Salt stress is among the most important abiotic stresses that are harmful to crop growth, development, and yield [4,5]. Therefore, improving plant salt tolerance to increase yield and quality is of great importance in plant breeding, food security, and economic sustainability [5].

To breed plants that can adapt to high salt stress, it is necessary to identify how plants tolerate this stress [6]. Under salt stress conditions, the morphological structures and ion flow in plants change to adapt to the hostile environment. For example, the expansion of root tips is reduced, and the stomata in young leaves are closed. The excretion of $Na^+$ from the roots increases, and the intake of $Na^+$ in the leaves is restricted. $Na^+$, $K^+$, and $Cl^-$ ions in tissue vacuoles are compartmentalized and transported to different tissues of the aerial

parts of the plant [7–9]. Calcium ($Ca^{2+}$) is the central regulator of physiological responses to stress [10,11]. The calcineurin B-like (CBL)-interacting protein kinase (CIPK) pathway and the salt overly sensitive (SOS) pathway are plant-specific $Ca^{2+}$ sensor-relaying pathways involved in the adaptation process of plants to salt stress [12–17]. Stresses induce changes in the concentration of intracellular $Ca^{2+}$, and CBL proteins capture these changes and activate their target genes, *CIPKs*, to exclude or compartmentalize $Na^+$ and maintain the $Na^+/K^+$ balance in the cytoplasm [18]. SOS3 (CBL4) cooperates with SOS2 (CIPK24) to transmit calcium signals to activate SOS1 on the plasma membrane via phosphorylation [19,20]. The activated SOS1 then extrudes $Na^+$ into the extracellular space to escape salt toxicosis. The CBL1/CBL9-CIPK23 complexes regulate $K^+$ in plants by regulating the $K^+$ channel [21]. This indicates the critical role of the CBL-CIPK network in salt detoxification.

*Populus euphratica* Oliv. is a typical stress-resistant tree that is widely planted in desert- and drought-endemic areas [22]. It is highly tolerant to abiotic stresses, such as salt, cold, drought, and alkaline conditions [23–25]. *P. euphratica* can survive under high NaCl conditions of up to 400 mM [26]. These characteristics make it a valuable natural genetic resource for salt-tolerant plant breeding. Due to the strong resistance of *P. euphratica* to stress, studying its CBL-CIPK pathway at the molecular level, as well as its regulation of the balance between $Na^+$ and $K^+$, holds substantial value. The CBL-CIPK members *PeCBL1*, *PeCBL6*, *PeCBL10*, *PeCIPK26* (*PeSOS2*), and *PeSOS1* have been cloned in *P. euphratica* [27,28]. However, no studies have been conducted regarding *PeSOS3*. Therefore, to investigate the resistance mechanism of *P. euphratica*, we cloned the *PeCBL4* (*PeSOS3*) gene and analyzed its subcellular localization and interacting proteins. Then, *PeCBL4* was transformed into *Arabidopsis thaliana* mutants (*sos3*) to characterize its function under high salt stress and low $K^+$ stress.

## 2. Materials and Methods

### 2.1. Plant Materials

Two-year-old seedlings of *P. euphratica* were planted in 10 L pots in a greenhouse for two months. During natural growth, water was supplied regularly every week. After two months, the roots, stems, and leaves were extracted separately from six uniformly developed seedlings. All plant materials were quickly frozen in liquid nitrogen and stored at −80 °C until use.

### 2.2. Sequence Analysis of PeCBL4

Total RNA was obtained according to a previously described method [29]. cDNA was synthesized and amplified using the following procedures: 94 °C for 1 min followed by 30 cycles of respective Tm for 50 s and 72 °C for 1 min using the TaKaRa Ex Taq Kit (TaKaRa). The primers of *peCBL4* were designed based on the *PtCBLs* of *Populus trichocarpa* [30] and are provided in Supplementary Materials Tables S1 and S2. The cDNA product was sequenced after insertion into the pMD18-T vector. The putative amino sequences of *PeCBL4* were aligned in DNAMAN software using the "multiple sequence alignment" method. The phylogenetic tree of the CBL family was constructed using the neighborhood joining Bootstrap method (Bootstrap N-J Tree) using Clustal X (1.8) software.

### 2.3. Subcellular Localization of PeCBL4

Subcellular localization of *PeCBL4* was performed on *Nicotiana benthamiana* as described previously [31,32]. Briefly, the coding sequences of *PeCBL4* were obtained using PCR (primers are displayed in Supplementary Materials Table S1) and cloned into the pBIB::GFP (GW) vector to generate pBIB::GFP-PeCBL4. The constructs were then inserted into *Agrobacterium tumefaciens* strain GV3101, and the GV3101 bacteria carrying the fusion construct pBIB::GFP-PeCBL4 and p19 protein were infiltrated into five-to-six-week-old *N. benthamiana* leaves, respectively. The infected *N. benthamiana* plants were cultivated for two days, and the leaves were prepared for microscopic analysis under GFP fluorescence

and white light. Protoplasts from leaves were prepared as described in a previous study [31] for microscopic analysis of the fluorescence distribution.

### 2.4. Yeast Two-Hybrid Assay

A yeast two-hybrid assay was performed to determine the relationship between CBL and CIPK, according to previous studies [28,32]. The coding sequences of PeCBL and PeCIPK were obtained using PCR (primers are displayed in Supplementary Materials Table S1) and inserted into the pGBKT7 or pGADT7 vector, respectively. The pMD18-T constructs were used as templates for PCR reactions (for PeCBL4 [33]). The binding domain (BD) constructs carrying PeCBL and activation domain (AD) constructs carrying PeCIPK were inserted into the yeast two-hybrid reporter strain AH109 using the lithium acetate method. The AH109 strain was then transferred to 30 °C environment and cultured for 2–3 days on SC-Trp/Leu media plates. Finally, the AH109 strain containing vectors of AD and BD were inoculated onto an SC-Leu-Trp-His-Ade plate and cultured at 30 °C for three-to-five days and captured.

### 2.5. Bimolecular Fluorescence Complementation (BiFC) Assay

PeCBL4 was inserted into the pSPYCE(M) vector using Xba I and Sal I as restriction enzymes to generate the BiFC constructs. PeCIPK24, 25, 26, and 27 were introduced into the pSPYNE(R)173 vector using BamH I and Kpn I as restriction enzymes. Supplementary Materials Table S1 provided the primers in PCR. BiFC assays were performed as previously described [28,31]. Briefly, the coding sequences of *PeCBL4* and *PeCIPK24, 25, 26,* and *27* were obtained using PCR (primers are displayed in Supplementary Materials Table S1) and inserted into the pSPYCE(M) vector using XbaI and SalI as restriction enzymes or pSPYNE(R)173 vector using BamHI and KpnI as restriction enzymes, respectively. These constructs were then inserted into *A. tumefaciens* strain GV3101 and further infiltrated into *N. benthamiana* five-to-six-week-old leaves. Microscopic analysis of YFP fluorescence was conducted after four days.

### 2.6. Arabidopsis thaliana Transformation and Phenotype Identification

*A. tumefaciens* GV3101 carrying pBIB::GFP-PeCBL4 was transformed into *A. thaliana* mutant *sos3-1* via the floral dip method [27]. T1 transformants were seeded in the soil and sprayed with a 0.0002% ($v/v$) BASTA. The genotypes of the surviving plants were further determined using PCR. For phenotypic identification, transgenic homozygotes, corresponding mutants, and T3 seeds of wild-type (WT) plants were surface-sterilized in MS medium and stratified at 4 °C for two days to obtain uniform germination. After five days, the seedlings were transferred to an improved MS medium containing 100 MK + $NH_4H_2PO_4$ instead of $KH_2PO_4$ and $NH_4NO_3$ instead of $KNO_3$ or supplemented with 150 mM NaCl. Seeds were grown for seven days and subsequently analyzed the phenotype. The apical position of root was marked immediately after the transfer, and the root elongation was quantified with a ruler finally.

### 2.7. Determination of Na$^+$ and K$^+$ Content

To measure the Na$^+$ and K$^+$ content in the plants, $T_1$ transformants were allowed to grow on 1/2MS medium for four days. They were then transferred to MS media containing high levels of K (100, 50, or 20 μM K$^+$) for seven days or transferred to MS media with high salt levels (100, 120, or 150 mM Na$^+$) for 48 h. Then, seedlings were put in an envelope, dried at 80 °C for three-to-five days, and weighed. Then, 60 mg of the dried sample was ground into powder and nitrified overnight with concentrated nitric acid. Subsequently, 30% $H_2O_2$ was added and then placed at 140 °C for 4 h to evaporate $HNO_3$ and $H_2O_2$. The sample was then diluted with distilled water to a total volume of 25 mL and analyzed using a flame atomic absorption spectrophotometer (Spectr AA-220, Cal-L Enterprises, Chatsworth, CA, USA).

*2.8. Measurement of Na⁺ and K⁺ Flux Using the Noninvasive Micro-Test Technique (NMT)*

The net fluxes of $Na^+$ and $K^+$ were determined via NMT (Xuyue (Beijing, China) Sci. & Tech Co., Ltd., Applicable Electronics Inc (Fort Worth, TX, USA)., ScienceWares Inc (Falmouth, MA, USA)., and Younger USA Corp (Amherst, MA, USA).), which is a non-invasive technique for measuring the dynamic flow direction and rate of ions or molecules on material surfaces [34]. The *A. thaliana* seeds were placed at 4 °C for two days and then at 22 °C for six days. The seeds were then transferred to 1/2MS medium and allowed to grow for six days. Subsequently, they were transferred to 1/2MS medium containing 150 mM NaCl or 50 μM $K^+$ ($KH_2PO_4$ was replaced by $NH_4H_2PO_4$ and $KNO_3$ was partially replaced by $NH_4NO_3$) for one day. Then, the net $K^+$ and $Na^+$ fluxes were measured as described previously [32].

*2.9. Statistical Analysis*

Data are presented as mean ± standard deviation (SD) and were analyzed using SPSS 23.0 software. The ionic fluxes were calculated using Mageflux software (Xuyue (Beijing) Sci. & Tech Co., Ltd., Beijing, China). Differences among groups were calculated via one-way ANOVA using the LSD test. $p < 0.05$ indicated a significant difference.

## 3. Results

*3.1. Isolation and Sequence Analysis of PeCBL4*

We cloned a new CBL member from *P. euphratica* and named it *PeCBL4* (gene accession number: DQ907706). This gene is highly homologous to *AtCBL4* in *A. thaliana*. The open reading frame of this gene was 642 bp encoding 212 amino acid residues. Phylogenetic tree analysis of *PeCBL4* from *P. euphratica* with CBL members in *P. trichocarpa* and *A. thaliana* showed that *PeCBL4* and *PtCBL4* were clustered together and belonged to the same subgroup as *AtCBL4* (Figure 1A). This suggested that these three genes originated from the same ancestor. Through the alignment of the amino acid sequences of *PeCBL4*, *PtCBL4*, and *AtCBL4*, we found that their structures were extremely conserved. Only six amino acids were different between *PeCBL4* and *PtCBL4*. Similar to other CBL members, the *PeCBL4* amino acid sequence contained four EF hand structures, and these structures were conserved in length (Figure 1B). Taken together, by comparing the sequences of *PeCBL4* with the 10 CBL members of *A. thaliana*, we showed that *PeCBL4* is homologous to *AtCBL4* in *A. thaliana*.

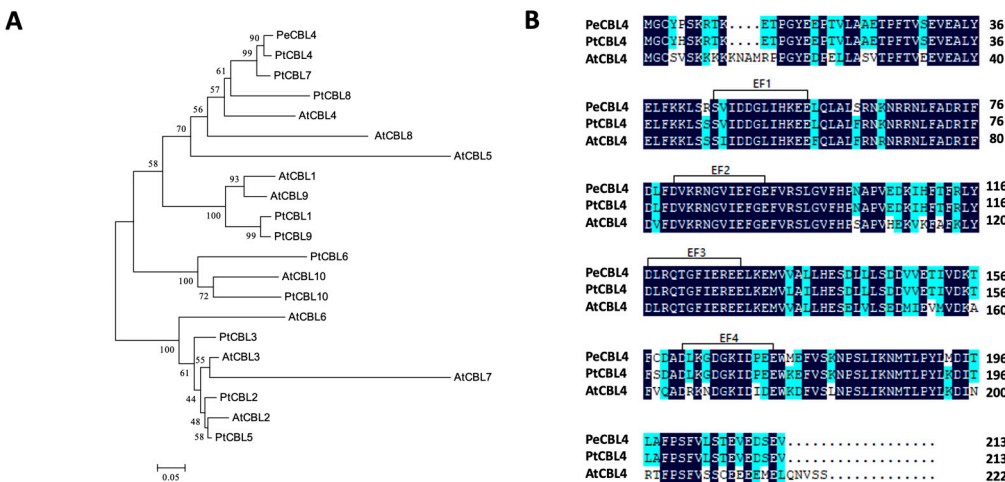

**Figure 1.** The identification of *PeCBL4* gene. (**A**) Phylogenetic tree analysis of *PeCBL4* from *Populus euphratica* with CBL members in *Populus trichocarpa* and *Arabidopsis thaliana.* (**B**) Alignment of the deduced amino acids *PeCBL4*, *PtCBL4*, and *AtCBL4.* The amino acids with dark blue backgrounds indicate full homology among *P. euphratica*, *P. trichocarpa*, and *A. thaliana*. The amino acids with light blue backgrounds indicate part homology.

### 3.2. Subcellular Localization of PeCBL4

To determine the specific part of the PeCBL4 protein functioning in the cell, a fusion expression vector of GFP (pBIB::GFP) and PeCBL4 was constructed. The fusion expression vector of pBIB::GFP-PeCBL4 was inserted into GV3101 and transiently expressed in epidermal cells and protoplasts of *Nicotiana tabacum.* As shown in Figure 2, the GFP signal in the cells transformed with the control plasmid pBIB::GFP was observed throughout the cells, while the GFP signal in the cells transformed with pBIB::GFP-PeCBL4 was localized to the plasma membrane.

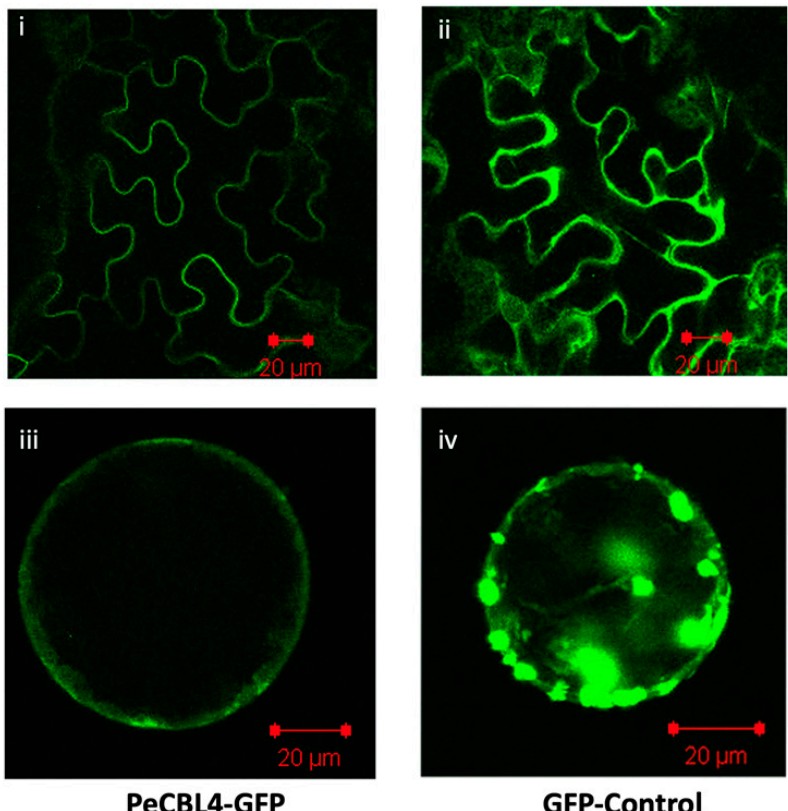

**Figure 2.** Subcellular localization analyses of PeCBL4. Subcellular localization of PeCBL4 in tobacco epidermal cells and protoplasts. (**i**) Tobacco epidermal cells expressing pBIB::GFP-PeCBL4 under green channel fluorescence. (**ii**) Tobacco epidermal cells expressing pBIB::GFP under green channel fluorescence. (**iii**) Protoplasts expressing pBIB::GFP-PeCBL4 under green channel fluorescence. (**iv**) Protoplasts expressing pBIB::GFP under green channel fluorescence. Bar = 20 µm.

### 3.3. Identification of PeCIPK Members Interacting with PeCBL4 via Yeast Two-Hybrid Assay

Yeast two-hybrid assays were performed to identify PeCIPK members that interacted with PeCBL4. An interaction between AD- and BD-fusion proteins resulted in the growth of yeast cells in a medium lacking adenine and histidine. As shown in Figure 3A, only cells containing both plasmids encoding AD-PeCIPK24 and BD-PeCBL4 or AD-PeCIPK26 and BD-PeCBL4 could grow in SC medium lacking adenine and histidine, indicating that PeCBL4 could only interact with PeCIPK24 (homologous to AtCIPK23) and PeCIPK26 (homologous to AtSOS2). Their interactions still existed when we exchanged the vectors of PeCBL4 and PeCIPK members (Figure 3B). These results indicated that the CBL-CIPK signaling network in *P. euphratica* may be more complex than that in *A. thaliana*.

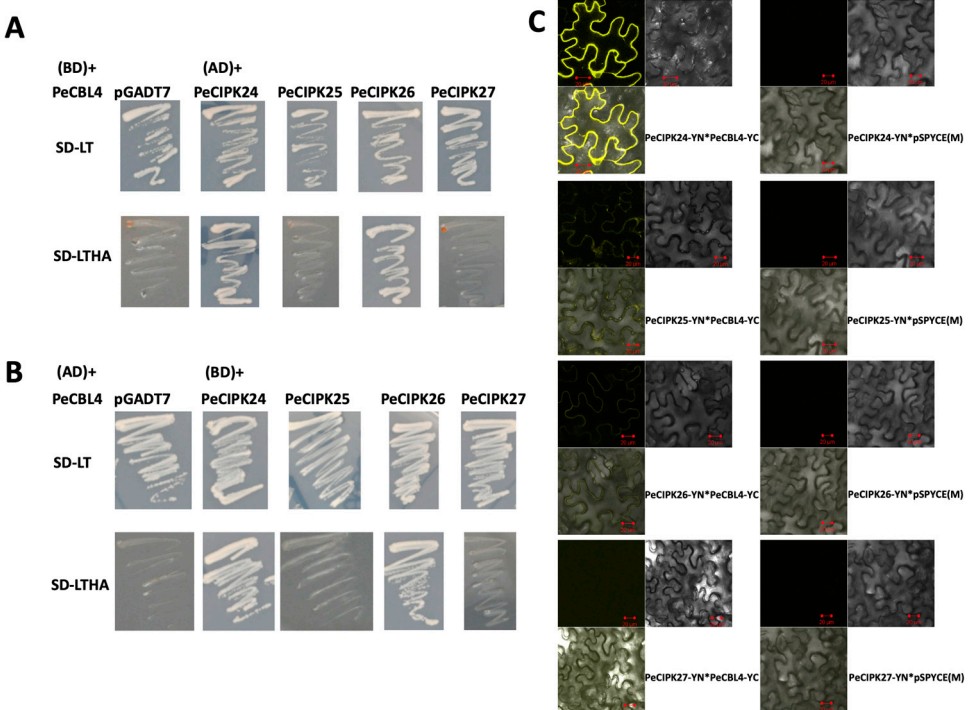

**Figure 3.** Identification of genes interacting with PeCBL4. (**A**) A yeast two-hybrid assay identified the PeCIPK members that interacted with PeCBL4. AH109 yeast cells containing plasmids encoding either AD-PeCIPK members of PeCIPK24, PeCIPK25, PeCIPK26, PeCIPK27, or BD-PeCBL4 were plated on non-selective (SD-LT) and selective (SD-LTHA) plates and incubated at 30 °C for three days. The blank pGADT7 served as the control. (**B**) A yeast two-hybrid assay identified the PeCIPK members that interacted with PeCBL4. AH109 yeast cells containing plasmids encoding either BD-PeCIPK members PeCIPK24, PeCIPK25, PeCIPK26, PeCIPK27, or AD-PeCBL4 were plated on non-selective (SD-LT) and selective (SD-LTHA) plates and incubated at 30 °C for three days. The blank pGBKT7 served as the control. (**C**) BiFC assays of PeCBL4 and PeCIPK members in vivo. The PeCBL4-YC with PeCIPKs-YN was coexpressed in *N. benthamiana* leaf cells and confocal images of YFP fluorescence in cells indicated the interaction. The co-expression of pSPYCE (M) with PeCIPKs-YN was used as a control. Bar = 20 μm.

*3.4. Identification of the Downstream Genes of PeCBL4 in P. euphratica via BiFC*

The yeast two-hybrid assay preliminarily indicated that PeCBL4 interacted with PeCIPK24 and PeCIPK26. However, we believe that there might be false positive results. The yeast two-hybrid assay result may have been caused by the heterogeneity between the plants and yeast. Therefore, we further performed BiFC to verify the results of the yeast two-hybrid assay. As shown in Figure 3C, YFP fluorescence was observed in *N. benthamiana* leaf cells transfected with both PeCBL4 and PeCIPK24, PeCBL4 and PeCIPK25, and PeCBL4 and PeCIPK26, indicating an interaction between PeCBL4 and PeCIPK24, PeCBL4 and PeCIPK25, and PeCBL4 and PeCIPK26. No fluorescence was observed in *N. benthamiana* leaf cells transfected with PeCBL4 and PeCIPK27. This result further clarifies that PeCBL4 can interact with PeCIPK24 and PeCIPK26, which is the same as the yeast two-hybrid assay. In addition, we found that the positions where they act in the cell were all on the plasma membrane, which is consistent with the results of the GFP signal in the subcellular localization of the PeCBL4 protein previously studied.

Unlike the yeast two-hybrid results, we found that there was also a signal response between PeCBL4 and PeCIPK25, although the fluorescence signal was weaker than that between PeCBL4 and PeCIPK24.

### 3.5. Constitutive Expression of PeCBL4 into the A. thaliana sos3 Mutant Functionally Complemented the sos3 Mutant and Enhanced Tolerance to Salt Stress

*AtCBL4* (*AtSOS3*) is involved in signal transduction under high salt stress. Therefore, we investigated whether *PeCBL4* is involved in regulating resistance to salt stress. *A. tumefaciens* GV3101 carrying pBIB::GFP-PeCBL4 was transformed into *A. thaliana* mutant *sos3* using the floral dip method. The growth status of these lines was examined on 1/2MS medium. The genotype was further confirmed via qRT-PCR (Figure 4A). As shown in Figure 4B,D, the root growth of the *sos3* mutant was significantly longer than that of the WT, while that of the *PeCBL4* transgenic *A. thaliana* plants was similar to that of the WT under low K$^+$ treatment. Under high salt conditions, the *PeCBL4* transgenic *A. thaliana* plants grew more lateral roots, and the leaves no longer produced albinism compared with mutant *sos3* plants (Figure 4C,D).

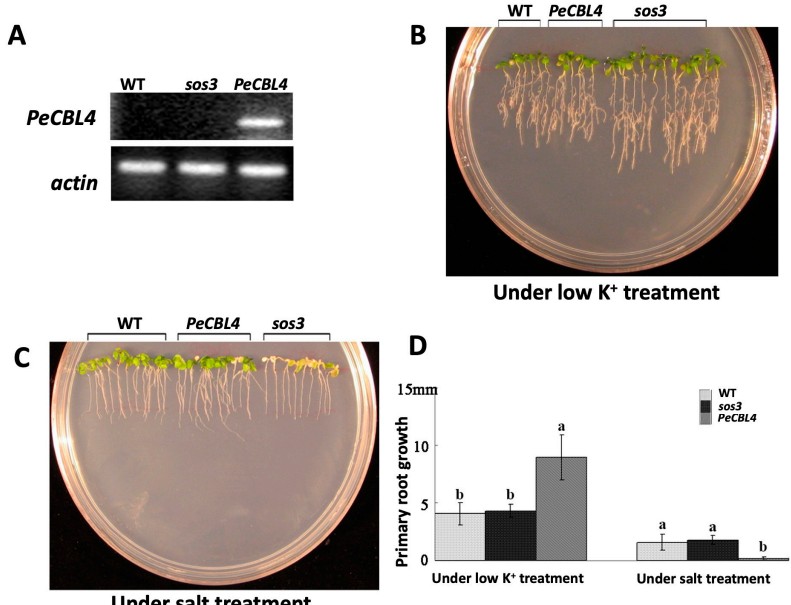

**Figure 4.** Phenotype analysis of transgenic plants of *A. thaliana* transformed with *PeCBL4* (cbl4 background) under low K$^+$ or high salt conditions. (**A**) Expression analysis of *PeCBL4* in wild-type (WT), *sos3* mutant, and *PeCBL4* transgenic *A. thaliana* plants. The *Arabidopsis* actin gene was used as a control. (**B**) Phenotype analysis of WT, *sos3* mutant, and *PeCBL4* transgenic plants under low K$^+$ stress. The root growth of the *sos3* mutant was significantly longer than that of the WT. The *PeCBL4* transgenic was similar to that of the WT under low K$^+$ treatment. (**C**) Phenotype analysis of WT, *sos3* mutant, and *PeCBL4* transgenic plants under high salt conditions. The *PeCBL4* transgenic plants grew more lateral roots, and the leaves no longer produced albinism compared with mutant *sos3* plants. (**D**) Comparative analysis of root growth and extension of WT, *sos3* mutant, and *PeCBL4* transgenic plants under low K$^+$ and high salt stress conditions. Groups labeled with the same letter indicate $p > 0.05$, while different letters indicate $p < 0.05$.

### 3.6. Constitutive Expression of PeCBL4 in A. thaliana sos3 Mutant Influenced the K$^+$ and Na$^+$ Content in Seedlings

Through phenotypic analysis, we found that *PeCBL4* transgenic plants complemented the phenotype of *sos3* mutants under high salt stress but showed negative regulation under low K$^+$ stress. To further validate these effects, we detected the K$^+$ and Na$^+$ contents of the WT, *PeCBL4* transgenic plants, and *sos3* mutant plants under different concentrations of K$^+$ or Na$^+$. As shown in Figure 4, the content of K$^+$ in the whole seedlings decreased with a decrease in the concentration of K$^+$ in the culture environment. The K$^+$ in the whole seedlings decreased from 50–55 mg/gdw under MS to 20–25 mg/gdw under treatment with 20 μM K$^+$ (Figure 5A). We also noted that the K$^+$ content in the whole seedlings was higher in the *sos3* mutants than in the WT and *PeCBL4* transgenic plants under all conditions

(Figure 5A). In addition, the amount of K$^+$ in the *sos3* mutants was less reduced compared with that of the WT and *PeCBL4* transgenic plants (Figure 5A). On the contrary, the Na$^+$ content in the whole seedlings gradually increased with a decrease in the concentration of K$^+$ treatment (Figure 5B). The increase in Na$^+$ content in the whole seedlings of the mutant was greater than that in the WT and *PeCBL4* transgenic plants (Figure 5B). Overall, the ability of *sos3* mutants to maintain K$^+$ content in plants is greater than that of WT and transgenic plants. These results indicated that *PeCBL4* transgenic plants exhibited a sensitive phenotype under low K$^+$ stress.

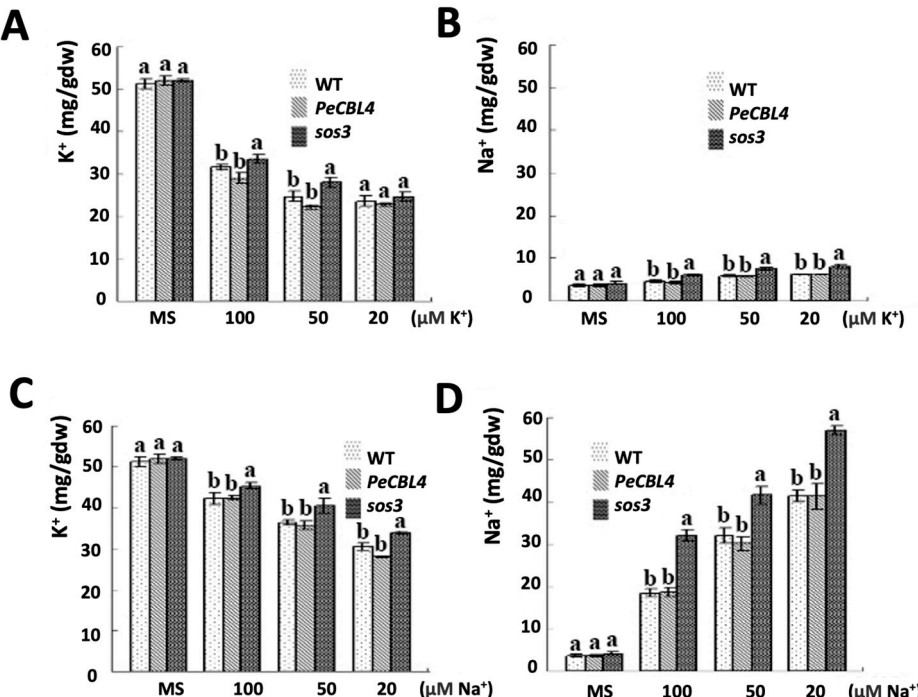

**Figure 5.** Comparison of K$^+$ and Na$^+$ content in WT, *PeCBL4*, and *sos3* seedlings under low K$^+$ or high salt stress. (**A**) The K$^+$ content in whole seedlings under MS or different concentrations of K$^+$. The K$^+$ content in the whole seedlings was higher in the *sos3* mutants than in the WT and *PeCBL4*. (**B**) The Na$^+$ content in whole seedlings under MS or different concentrations of K$^+$. The increase in Na$^+$ content in the whole seedlings of the mutant was greater than that in the WT and *PeCBL4*. (**C**) The K$^+$ content in whole seedlings under MS or different concentrations of Na$^+$. (**D**) The Na$^+$ content in whole seedlings under MS or different concentrations of Na$^+$. Results are presented as the mean of three independent replicates $\pm$ standard deviation. Statistical analysis was performed with ANOVA with LSD post hoc analysis. Groups labeled with the same letter indicate $p > 0.05$, while different letters indicate $p < 0.05$.

Under salt stress, the Na$^+$ content in the whole seedlings increased, while the K$^+$ content decreased with increasing Na$^+$ concentrations in all three types of seedlings (Figure 5C). Similar to the previous results, the decrease in K$^+$ in the *sos3* mutants was less than that of the WT and *PeCBL4* transgenic plants, and the increase in Na$^+$ content in the *sos3* mutants was greater than that of the WT and *PeCBL4* transgenic plants (Figure 5D). This indicated that *PeCBL4* improved the resistance of transgenic plants to salt stress.

*3.7. Constitutive Expression of PeCBL4 into the A. thaliana sos3 Mutant Influenced the K$^+$ and Na$^+$ Net Fluxes in the Roots*

To further test the selectivity for K$^+$ and Na$^+$ in different plants, we detected the efflux and influx of K$^+$ and Na$^+$ in the WT, *sos3* mutant, and *PeCBL4* (*PeSOS3*) transgenic plants under normal growth conditions, high Na$^+$ stress (150 mM NaCl treatment for 24 h), and low K$^+$ stress (50 μM K$^+$ treatment for 24 h).

As shown in Figure 6A, the $Na^+$ efflux capacity of all seedlings was 61–95 pmol·cm$^{-2}$·s$^{-1}$ under normal conditions. After high $Na^+$ treatment, the $Na^+$ efflux capacities of WT seedlings increased to 69–220 pmol·cm$^{-2}$·s$^{-1}$, while the net $Na^+$ flux of the *sos3* mutant significantly decreased to 23 pmol·cm$^{-2}$·s$^{-1}$ ($p < 0.05$). However, the net $Na^+$ flux of PeCBL4 recovered to similar levels as those of the WT ($p > 0.05$). These results indicated that the $Na^+$ efflux capacities of seedlings were enhanced under salt stress, except for the *sos3* mutants. Moreover, *PeCBL4* recovered the loss of function and complemented the phenotypes of their corresponding mutants. Under low $K^+$ treatment, the WT and *sos3* mutants showed absorption of $Na^+$, whereas the *PeCBL4* transgenic plants exhibited $Na^+$ efflux (Figure 6A). These results indicate that *PeCBL4* participated in the regulation of net $Na^+$ flux under high $Na^+$ and low $K^+$ stress.

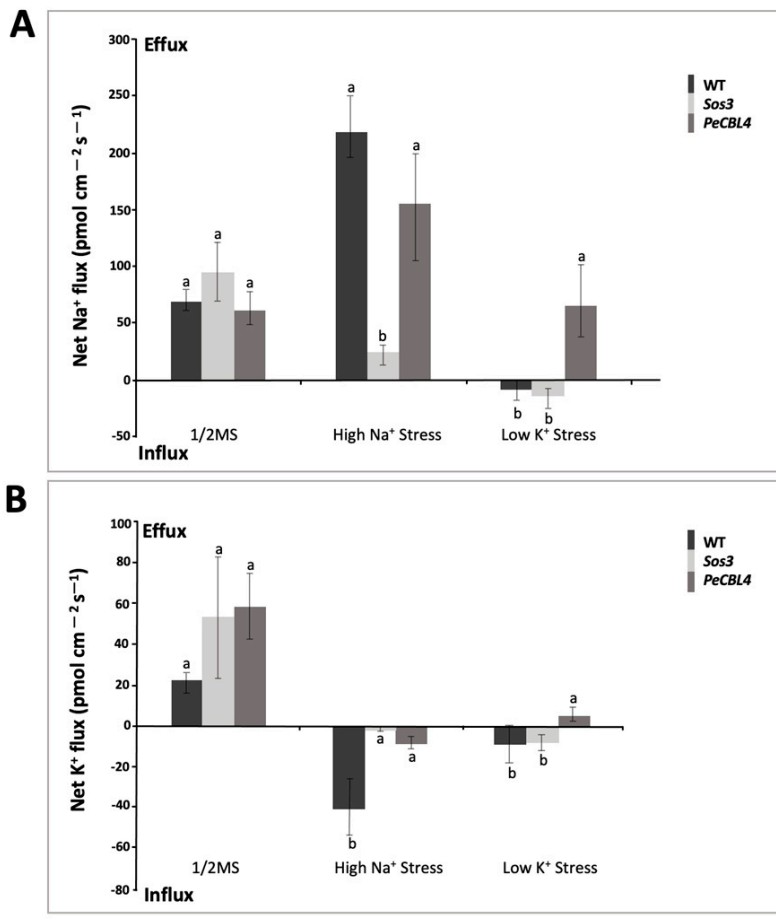

**Figure 6.** Net $Na^+$ and $K^+$ fluxes in the roots of plants. (**A**) Net $Na^+$ fluxes in the roots of plants under 1/2MS medium, high salt stress (150 mM NaCl for one day), and low $K^+$ stress (50 μM $K^+$ treatment for one day). (**B**) Net $K^+$ fluxes in the roots of plants under 1/2MS medium, high salt stress (150 mM NaCl for one day), and low $K^+$ stress (50 μM $K^+$ treatment for one day). Differences among groups were calculated via one-way ANOVA using the LSD test. Groups labeled with the same letter indicate $p > 0.05$, while different letters indicate $p < 0.05$.

Under normal conditions, all seedlings showed $K^+$ efflux at 23–59 pmol·cm$^{-2}$·s$^{-1}$ (Figure 6B). After treatment with high levels of salt, all three types of plants showed $K^+$ influx. In addition, we found that the influx capacity to $K^+$ of *PeCBL4* plants was greater than that of *sos3* mutants, but the difference was not statistically significant (Figure 6B). After low $K^+$ treatment, WT and *sos3* mutants showed absorption of $K^+$, while PeCBL4 transgenic plants showed $K^+$ efflux ($p < 0.05$; Figure 6B). This means that *PeCBL4* plays a negative role in maintaining $K^+$ content under low $K^+$ conditions.

## 4. Conclusions

*P. euphratica* is a typical stress-resistant tree that provides a valuable natural genetic resource for salt-tolerant plant breeding. The CBL-CIPK network plays an important role in regulating plant responses to abiotic stresses [14,16,35–38]. A number of CBL and CIPK family members have been cloned in many plant species. For example, 10 CBLs and 26 CIPKs have been found in *A. thaliana*, 24 CBLs and 79 CIPKs in *Triticum aestivum*, 8 CBLs and 26 CIPKs in *Manihot esculenta*, 10 CBLs and 34 CIPKs in *Oryza sativa*, 7 CBLs and 23 CIPKs in *Brassica napus*, 9 CBLs and 26 CIPKs in *Capsicum annuum*, and 10 CBLs and 27 CIPKs in *P. trichocarpa* [36,37,39–44]. Our previous studies identified and characterized several members in the CBL-CIPK network in *P. euphratica*, including *PeCBL1*, *PeCBL6*, *PeCBL10*, *PeCIPK26* (*PeSOS2*), and *PeSOS1* [27,28]. In this study, the PeCBL4 (PeSOS3) gene was cloned to characterize its function under high salt stress and low $K^+$ stress.

Phylogenetic tree analysis showed that *PeCBL4* was clustered together with *PtCBL4* and belonged to the same subgroup as *AtCBL4*. This result indicated that *PeCBL4* is highly homologous to *AtCBL4* in *A. thaliana*. Amino acid alignment results suggested that the structure of *PeCBL4* was highly conserved with *PtCBL4* and *AtCBL4* and contained four EF hands. The amino acid sequences encoded by the CBL family members are conservative, and they all have four EF hands to capture $Ca^{2+}$ [45,46]. Subcellular localization indicated that PeCBL4 was expressed on the plasma membrane. This result was consistent with the location of CBLs in other species. In most plants, CBLs recruited CIPKs to the plasma membrane or tonoplast to form a functional complex. The CBL1/CBL9-CIPK23 complexes are reported to be localized to the plasma membrane and regulate $K^+$ in stomatal guard cells and roots [21]. The $Na^+/H^+$ exchange contributed to cellular $Na^+$ homeostasis in the plasma membrane of plant cells [19]. In *A. thaliana*, AtSOS2 and AtSOS3 capture salt signals and further activate AtSOS1, which confers salt tolerance [19,47,48].

CBLs function in salt tolerance by forming complexes with CIPKs. Each CBL can interact with multiple CIPKs after capturing the $Ca^{2+}$ signal in the cellular environment [33]. To identify the PeCIPKs that interact with PeCBL4, we performed yeast two-hybrid assays and BiFC. Results showed that PeCBL4 interacted with PeCIPK24 and PeCIPK26 (PeSOS2). The interaction between CBL4/SOS3 and CIPK24/SOS2 has frequently been reported in other species. In *A. thaliana*, AtCBL4 binds to AtCIPK24 to activate SOS1 and function in excluding $Na^+$ from the cytoplasm [49]. This result indicated that there is also a conservative SOS signaling pathway in *P. euphratica*. Although the yeast-two hybrid assay and BiFC were both used to detect interactions between two proteins, their results may be different. This is partly because BiFC detects indirect interactions that might be overlooked in the yeast-two hybrid assay [50]. Moreover, the PeCBL4 was found operating on the plasma membrane, which is consistent with the results of the GFP fluorescence signal in the subcellular localization. The results further confirmed that the location of the interaction between CBL and CIPK in the cell was determined via CBL.

At present, it is difficult to elucidate the gene functions and their regulation in woody plants. Therefore, the use of model plants to study the regulation mechanisms of forest gene function has become an important method. To further characterize the function of *PeCBL4*, we transformed this gene to *A. thaliana* and successfully obtained homozygous transgenic *A. thaliana* plants. In *A. thaliana*, *AtCBL4* (*SOS3*) and *AtCIPK24* (*SOS2*) are involved in signal transduction under high salt stress [49]. We found that PeCBL4 interacted with PeCIPK24 and PeCIPK26 in both yeast two-hybrid assays and BiFC. We considered whether both CBL4-CIPK24 and CBL4-CIPK26 participated in signal transduction under adversity stress. Results indicated that the overexpression of *PeCBL4* in the *A. thaliana* mutant *sos3* complemented the phenotype of salt stress sensitivity caused by the deletion of the *CBL4* gene. This effect was accomplished via PeCBL4/SOS3-PeCIPK26/SOS2. Previously, we found that PeCBL1 interacted with PeCIPK24, PeCIPK25, and PeCIPK26 to regulate $Na^+/K^+$ homeostasis [32]. The overexpression of PeCBL4 in the sos3 mutant inevitably breaks the interaction balance between PeCBL1-PeCIPK24 and PeCIPK25, thus generating the phenotypic characteristics of transgenic plants sensitive to low $K^+$ stress.

The $Na^+$ and $K^+$ flux under low potassium stress and high salt stress were further detected using NMT. NMT is a relatively new method developed in the late 20th century. It can be used to obtain dynamic information on specific ionic activities on material surfaces without damaging the materials [34]. It has been used to detect multiple ions, such as $Al^{3+}$, $Cd^{2+}$, $Ca^{2+}$, $Mg^{2+}$, $H^+$, $K^+$, $Cl^-$, $NO^-$, and $O_2$ [34]. In this study, as well as in previous studies, we found that the efflux of $Na^+$ decreased in *sos3* and *sos2* mutants under high salt stress, while that in *PeCBL4/SOS3* and *PeCIPK26/SOS2* increased. The results suggested that the efflux capacities of seedlings to $Na^+$ were enhanced under salt stress, except for *sos3* and *sos2* mutants. This is consistent with the results of a previous study that indicated that the regulation of *SOS3-SOS2* is involved in the regulation of the salt signaling pathway in *A. thaliana* [49]. Moreover, *PeCBL4/SOS3* and *PeCIPK26/SOS2* recovered the loss of function and complemented the phenotypes of their corresponding mutants. This result indicated that PeCBL4/SOS3-PeCIPK26/SOS2 is also involved in the regulation of salt signaling.

In addition, the WT, *PeCBL1* transgenic plants, and *sos3* mutants showed absorption of $K^+$, whereas *cbl1/9* mutants and *PeCBL4* transgenic plants showed efflux of $K^+$ after low $K^+$ treatment [6]. Meanwhile, *PeCBL4* plays a negative role in maintaining $K^+$ content. The CBL1-CIPK23 pathway is involved in the regulation of $K^+$ transport under low $K^+$ stress [15,16,51]. Our results showed that PeCBL1 and PeCBL4 also affected the $K^+$ content in plants.

In conclusion, we cloned a new CBL member from *P. euphratica*, PeCBL4, and found that it could interact with PeCIPK24 and PeCIPK26 and function in regulating the $Na^+/K^+$ balance under low $K^+$ stress and high salt stress.

**Supplementary Materials:** The following supporting information can be downloaded at: https://www.mdpi.com/article/10.3390/f14071504/s1, Table S1: The primers for creating constructs; Table S2: GenBank accession numbers of *Populus* genes.

**Author Contributions:** F.L. conceived the study; M.Q., Q.S. and Z.C. collected and synthesized the data; N.C., H.Z., F.L. and Y.A. drafted the manuscript. All authors have read and agreed to the published version of the manuscript.

**Funding:** This work was supported by the National Natural Science Foundation of China (Grant No. 31700531), Zhejiang students' technology and innovation program, and Zhejiang A&F University student research training project.

**Data Availability Statement:** All data are available upon reasonable request.

**Acknowledgments:** We are grateful to Xinli Xia (College of Biological Sciences and Technology, Beijing Forestry University, Beijing, China) for her kind help and valuable suggestions.

**Conflicts of Interest:** The authors declare that the research was conducted in the absence of any commercial or financial relationships that could be construed as potential conflicts of interest.

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
