# Peer review of "Functional Characterization of a New Salt Stress Response Gene, PeCBL4, in Populus euphratica Oliv"

_forests, doi:10.3390/f14071504_

Round 1

Reviewer 1 Report

In this manuscript, the authors have done a complete study related to CBL4 gene in Populus euphratica. In my opinion, it has a great potential. The English of text is acceptable. I just have some comments:  

-          Lines 14, 15, etc. Gene names must be provided in italic. Please check the whole text.

-          Recently, Arab et al. 2023 (https://www.mdpi.com/2073-4425/14/3/753) done a comprehensive analysis of calcium sensor families, this manuscript can be used in INTRODUCTION.

-          Material and Methods section is well written. Just add the version of SPSS software (line 164).

-          The resolution of the images is low. They should be improved.

Author Response

Response to Reviewer #1

Thank you for your valuable comments and suggestions.

[Q1] Lines 14, 15, etc. Gene names must be provided in italic. Please check the whole text.

[A1] We have revised gene names in italic in the manuscript.

[Q2] Recently, Arab et al. 2023 (https://www.mdpi.com/2073-4425/14/3/753) done a comprehensive analysis of calcium sensor families, this manuscript can be used in INTRODUCTION.

[A2] This is a good manuscript, and we have added it to the INTRODUCTION.

[Q3] Material and Methods section is well written. Just add the version of SPSS software (Line 164).

[A3] We have added “SPSS 23.0” in Line 164.

[Q4] The resolution of the images is low. They should be improved.

[A4] We have updated all images.

Reviewer 2 Report

In the present manuscript: "Functional characterization of a new salt stress response gene, PeCBL4, in Populus euphratica", the authors conducted an interesting study to understand the role of PeCBL4 gene in the resistance mechanism of P. euphratica.

The work it is well presented, however, I think it is necessary to implement the manuscript with more recent references. In addition, the figures should be improved, they are unclear.

Author Response

Response to Reviewer #2

Thank you for your suggestion.

[Q1] The work it is well presented, however, I think it is necessary to implement the manuscript with more recent references.

[A1] We have added some recent references to the INTRODUCTION. eg. Xiao et al., 2022; Mao et al., 2022; Arab et al., 2023…

[Q2] In addition, the figures should be improved, they are unclear.

[A2] We have updated all images, please check the manuscript.

Reviewer 3 Report

Dear authors,

it was a pleasure to review such a good and interesting manuscript. Interesting topic, but during reading and when I finished reading and revision, I did not feel it is complete. I recommend putting some guidelines for using these results in practice. I would like to see a chapter in the form of recommendations for use. What should be used with what to get the best results.

Author Response

Response to Reviewer #3

Thank you for your valuable comments and suggestions. Responses can be found in PDF below.
